# A radioligand binding assay for the insulin-like growth factor 2 receptor

Pavlo Potalitsyn[1,2], Irena Selicharová[1], Kryštof Sršeň[1], Jelena Radosavljević[1¤], Aleš Marek[1], Kateřina Nováková[1], Jiří Jiráček[1], Lenka Žáková[1]*

**1** Institute of Organic Chemistry and Biochemistry, The Czech Academy of Sciences, Prague, Czech Republic, **2** Department of Biochemistry, Faculty of Science, Charles University, Prague, Czech Republic

¤ Current address: Faculty of Chemistry, Department of Biochemistry, University of Belgrade, Belgrade, Serbia

* zakova@uochb.cas.cz

## Abstract

Insulin-like growth factors 2 and 1 (IGF2 and IGF1) and insulin are closely related hormones that are responsible for the regulation of metabolic homeostasis, development and growth of the organism. Physiological functions of insulin and IGF1 are relatively well-studied, but information about the role of IGF2 in the body is still sparse. Recent discoveries called attention to emerging functions of IGF2 in the brain, where it could be involved in processes of learning and memory consolidation. It was also proposed that these functions could be mediated by the receptor for IGF2 (IGF2R). Nevertheless, little is known about the mechanism of signal transduction through this receptor. Here we produced His-tagged domain 11 (D11), an IGF2-binding element of IGF2R; we immobilized it on the solid support through a well-defined sandwich, consisting of neutravidin, biotin and synthetic anti-His-tag antibodies. Next, we prepared specifically radiolabeled [$^{125}$I]-monoiodotyrosyl-Tyr2-IGF2 and optimized a sensitive and robust competitive radioligand binding assay for determination of the nanomolar binding affinities of hormones for D11 of IGF2. The assay will be helpful for the characterization of new IGF2 mutants to study the functions of IGF2R and the development of new compounds for the treatment of neurological disorders.

## Introduction

Insulin-like growth factor 2 (IGF2) is a 7.5 kDa mitogenic peptide hormone expressed predominantly in liver, but by other tissues as well [1]. IGF2 shares 50 % sequence homology and similar 3-D organization with IGF1 and insulin. IGF2 is a major regulator of cell growth, survival, migration and differentiation, especially during fetal development [2, 3]. However, in spite of recent significant improvement in the knowledge about the physiological and pathological roles of IGF2, there is still a lack of information about the full spectrum of its functions in the organism. IGF2 is clearly much less studied than IGF1 and insulin. The significance of IGF2 is supported by its high level in the serum of human adults, which is three times higher than the concentration of IGF1 and much higher than the concentration of insulin [4–6]. IGF2 gene expression is maternally imprinted and is tightly regulated [7]. It is known that

**Data Availability Statement:** All relevant data are within the manuscript and its Supporting Information files.

**Funding:** LZ: the Czech Grant Agency (Grant 19-14069S), https://gacr.cz/en/, The funders had no role in study design, data collection and analysis, decision to publish, or preparation of the manuscript JJ: Medical Research Council (Grant MR/R009066/1), https://mrc.ukri.org/, The funders had no role in study design, data collection and analysis, decision to publish, or preparation of the manuscript JJ: the European Regional Development Fund; OP RDE; Project: "Chemical biology for drugging undruggable targets (ChemBioDrug)" (No. CZ.02.1.01/0.0/0.0/16_019/0000729), https://opvvv.msmt.cz/, The funders had no role in study design, data collection and analysis, decision to publish, or preparation of the manuscript Institutional support was provided by projects RVO 61388963 (for the Institute of Organic Chemistry and Biochemistry), https://www.uochb.cz/en, The funders had no role in study design, data collection and analysis, decision to publish, or preparation of the manuscript

**Competing interests:** The authors have declared that no competing interests exist.

elevated IGF2 concentration in serum is associated with an increased risk of developing various cancers, including colorectal, breast, prostate and lung [3, 5, 6]. On the other hand, there is growing evidence about the positive role of IGF2 in learning and memory consolidation in adult rats [8–10]. IGF2 is expressed in the brain, during both development and adulthood, and its concentration declines with ageing [11]. Administration of recombinant IGF2 led to better learning and memory reactivation and significantly enhanced several types of hippocampal-dependent memories [6, 9, 12]. Moreover, increased IGF2 levels in the hippocampus improved the cognitive, cellular and synaptic functions of Alzheimer model mice [13]. All these findings suggest IGF2 as a promising target for clinical use in the treatment of neurodegenerative diseases.

The mitogenic effects of IGF2 are thought to be mediated primarily through IGF1R and insulin receptor isoform A (IR-A) [9], that are both tyrosine kinases transmembrane receptors [14]. IGF2's bioavailability is also regulated by six IGF-binding proteins (IGFBP1-6) [15]. Moreover, IGF2 also binds to the IGF2 receptor (IGF2R), for which it has the highest binding affinity among all the aforementioned receptors [16]. IGF2R is structurally distinct from tyrosine kinases, but its intrinsic catalytic activity has not yet been clearly defined (see below). IGF2R, also known as the mannose-6-phosphate (M6P) receptor, is found ubiquitously in human tissues as a 275–300 kDa full-length membrane-spanning protein with a small cytoplasmic tail. However, a truncated soluble form of the receptor is present in the circulation as well [17]. IGF2R is a single transmembrane glycoprotein, composed of a short C-terminal cytoplasmic domain connected via a single transmembrane domain to a large extracellular region consisting of 15 homologous repeats (domains display 16 %-38 % identity) [18, 19]. All domains share a similar topology, consisting of a flattened β-barrel, composed of two four-stranded antiparallel β-sheets held together by four conserved disulfide bonds. Domains 3, 5, and 9 have been shown to bind M6P-containing ligands [20], while IGF2 binding has been localized to domain 11 (D11). Domain 13 (D13) was proposed to enhance binding by reducing the $k_{off}$ rate of IGF2:D11-IGF2R interaction [21, 22]. The crystal and NMR structures of D11 alone or of domains D11-D14 [23–26] have already been solved. Furthermore, the complexes of IGF2 with D11 or D11-D13 were solved by NMR or X-ray crystallography as well [24–26] (Fig 1A). Besides IGF2 hormone and M6P-containing ligands, IGF2R binds, albeit weakly (in $10^{-6}$–$10^{-8}$ M range), a large number of ligands, including lysosomal enzymes, transforming growth factor-β (TGF-β), granzyme B, plasminogen, glycosylated leukemia inhibitory factor and retinoids [27, 28]. The receptor continuously circulates between the cell surface and the trans-Golgi network *via* clathrin-coated vesicles [29], mediating the trafficking of M6P-containing lysosomal enzymes from the trans-Golgi network to the lysosomes. Approximately 90 % of membrane-bound IGF2R is normally found within the cell; the remaining receptor is present at the cell surface, where its extracellular domain can exist in both monomeric and dimeric form [30]. IGF2R, often referred to as a tumor suppressor, is paternally imprinted [3]. The loss of heterozygosity of IGF2R can lead to development of certain types of tumors (e.g. lung, breast, or liver cancer) [31], and in addition, knockdown of IGF2R induces apoptosis in hemangioma cells [32].

In the past, the role of IGF2R was thought to consist only of the clearance of free IGF2 from the circulation, together with its established role in the internalization of lysosomal trafficking [2, 3, 34]. However, some studies also provided evidence for the role of IGF2R in IGF2 signaling, although it is less clear whether IGF2R itself is able to initiate transmembrane signaling [35–38]. Now it seems that the action of IGF2 via IGF2R employs a heterotrimeric G-protein-dependent mechanism to activate the ERK1/2 pathway [39]. It was also confirmed that not IGF1R, but IGF2R is a target receptor for IGF2 during memory consolidation and enhancement and neurogenesis [8–10, 12, 13]. Thus, the IGF2/IGF2R axis is an interesting candidate

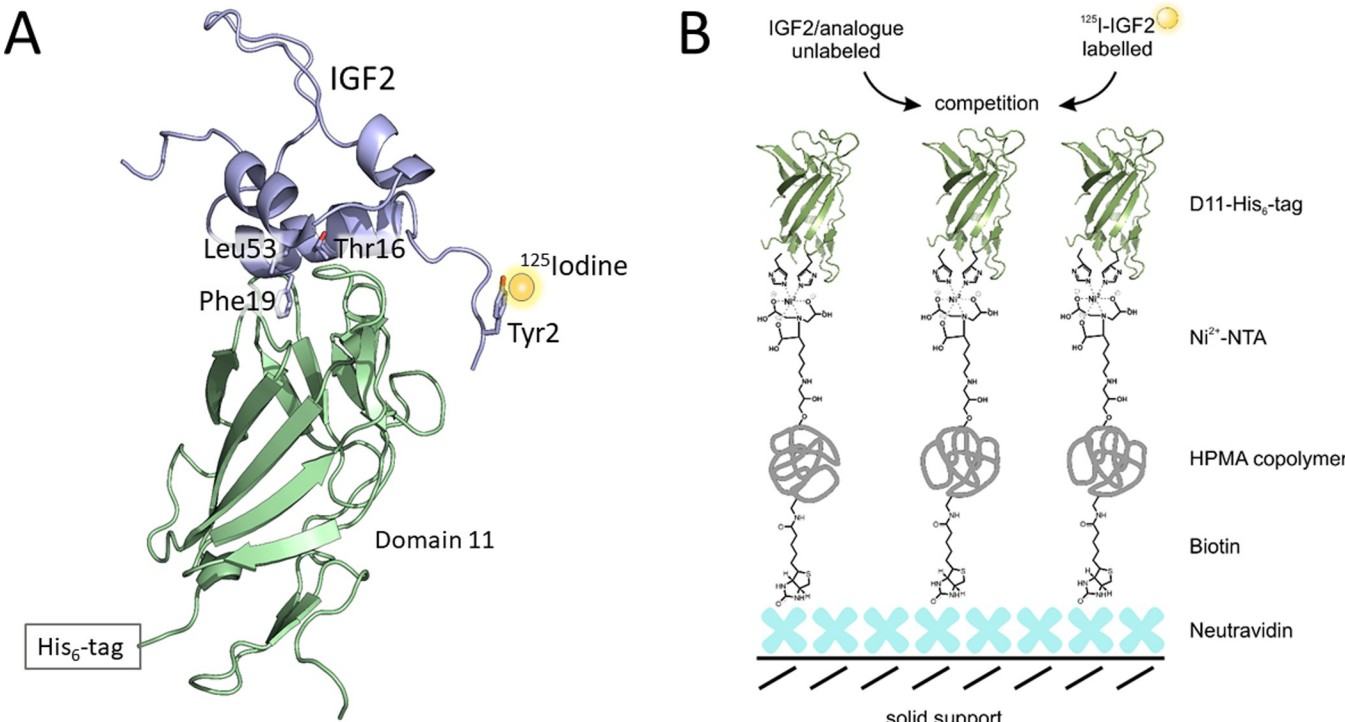

**Fig 1.** (A) Interaction of [$^{125}$I]-monoiodotyrosyl-Tyr2-IGF2 (in violet with a $^{125}$I atom shown as a yellow sphere) with His-tagged D11 of IGF-2R (in green). The side-chains of IGF2 residues involved in D11 binding are shown as sticks. Created based on 2L29.pdb structure [26]. (B) Schematic representation of the principle of IGF2:D11 radioligand binding assay developed in this study. iBodies® [33], consisting of N-(2-hydroxypropyl) methacrylamine (HPMA) copolymer bearing covalently attached biotin molecules and Ni$^{2+}$-chelating nitrilotriacetic acid (NTA) molecule, are immobilized on solid support (polystyrene wells) coated with neutravidin. Recombinant His-tagged D11 is immobilized in wells by binding of its His-tag to NTA-ligated Ni$^{2+}$ atom of iBodies®. Binding affinity of non-labeled ligands is determined through competition for D11 with [$^{125}$I]-IGF2 radiotracer.

for potential clinical interventions. Analogs of IGF2, with selective binding to IGF2R and suppressed affinity for "mitogenic" IGF1R and IR-A, could be important in the treatment of neurodegenerative diseases.

The determination of binding affinities of hormones to their relevant receptors plays a pivotal role in the design of new analogs with altered properties for structure-activity studies with receptors or for therapeutic applications. The highly sensitive binding assays employing $^{125}$I-labeled hormones for unequivocal determination of binding affinities of the hormones toward both isoforms of insulin receptor as well as toward IGF1 receptor have been established [40] and are routinely used in our studies [41–48].

At present, a sensitive radioligand competition binding assay for IGF2R is not easily accessible. This is partly because radiolabeled IGF2 is not currently available. In the early '90s, when $^{125}$I-IGF2 was available [49], laboratories doing IGF2R binding assays used partially purified rat placental membrane-derived IGF2R receptors [50], or IGF2R overexpressing baby hamster kidney cells [51, 52]. Other groups used fragments of IGF2R in isothermal titration calorimetry (ITC) measurements [53] or surface plasmon resonance (SPR) experiments [16, 26, 53, 54]. However, because ITC and SPR methods suffer from high material consumption and reduced sensitivity, a robust, unambiguous and sensitive radioligand binding assay for IGF2R binding could still represent the optimal solution.

Thus, here we developed a new and straightforward methodology based on a competition binding assay between a "cold" ligand and specifically $^{125}$I-radiolabeled IGF2 for D11 protein immobilized on a solid surface through iBodies® [33], an artificial polymer that can

specifically recognize histidine tags in proteins. The principle of our assay is schematically illustrated in Fig 1B. This setup and methodology have enabled the determination of affinities of IGF2 analogues for D11 as the main IGF2R domain interacting with IGF2. Thus, we developed a competition radioligand binding assay that allowed determination of the binding affinities of native IGF2, more potent [Leu19]-IGF2 analog and poorly binding IGF1. Our new methodology provided results that are fully compatible with previous studies on IGF2:IGF2R interaction and that could be useful for routine, sensitive and specific determination of the binding affinities of different ligands for IGF2R.

## Methods

Human IGF2 was purchased from GroPep Bioreagents (Adelaide, Australia) and human IGF1 was purchased from Sigma-Aldrich.

### Recombinant expression of Leu19-IGF2 analog

Leu19-IGF2 analog was produced, according to our previously published protocols [48, 55]. Briefly, Leu19-IGF2 was cloned into a modified pRSFDuet-1 expression vector as the fusion with an N-terminally His6 tagged-GB1 protein and TEV protease cleavage site. The construct was transformed to *E. coli* BL21(DE3) and purified and characterized as described previously. The successfully produced analog was purified by RP-HPLC and its purity was higher than 95 % (controlled by RP-HPLC analyses and HR-MS spectra).

### Recombinant expression and production of D11 domain of IGF2R

Domain 11 of human IGF2R (Uniprot code P11717, amino acids 1508–1651; D11) with six histidines (His6-tag) at the C-terminus of the sequence was cloned into pET24a expression vector. The construct was transformed into *E.coli* BL21(λDE3). Fermentation was done at 37 ˚C to reach optical density ∼1.0 at 600 nm, followed by an induction with 1 mM IPTG (isopropyl β-d-1-thiogalactopyranoside) and cultured for another 4 h. Cells were harvested by a centrifugation for 20 min at 6 000 × g at 4 ˚C. The supernatant was discarded and the cell pellet stored at -70 ˚C. Cells were resuspended in 50 mM Tris-HCl pH 8.7, 50 mM NaCl, 5 mM EDTA, 0.1 mM PMSF and lysed by Emulsiflex®. Cell lysate was centrifuged at 20 000 × g for 20 min. The supernatant was discarded and the pellet dissolved in the same buffer supplemented with 0.1 % Triton-X100. The same procedure was performed once more, but using a buffer without Triton-X100, and the pellet was sonicated on ice before centrifugation. The pellet obtained after the last centrifugation was stored at -20˚C. Inclusion bodies were dissolved in reducing and denaturing buffer (6 M guanidine-HCl, 50 mM Tris-HCl pH 8.7, 100 mM NaCl, 10 mM EDTA, 10 mM DTT) [24]. The resolubilized protein was incubated for 2 h at 4 ˚C with slow stirring. The fully denatured and reduced protein was dialyzed against 200 mM Tris-HCl pH 8.7, 10 mM EDTA, 1 M arginine, 0.1 mM PMSF, 6.5 mM cysteamine and 3.7 mM cystamine [23] for 48 h at 4 ˚ C with 2 buffer exchanges. The solution of the protein was centrifuged and chromatographed on a Superdex 75p 16/600 column. The collected protein was purified by RP-HPLC (Vydac C4 column) and the product was lyophilized and stored at -20 ˚C. The molecular weight was confirmed by high-resolution electrospray ionization mass spectrometry (LTQ Orbitrap XL, Thermo Fisher Scientific, Waltham, MA). The purity was checked by SDS-PAGE and RP-HPLC. The resulting D11 protein had more than 98 % purity (HPLC).

## Preparation of radiolabeled [125I]-monoiodotyrosyl-Tyr2-IGF2

Radiolabeled [125I]-monoiodotyrosyl-Tyr2-IGF2 was prepared via radioiodination of Tyr2 of IGF2 with 125I (Na[125I], product code: I-RB-41, 2 mCi, IZOTOP, Hungary), using the IODO-GEN™ system (Pierce) [56]. An Eppendorf PP tube (1.5 mL) pre-coated (*in house*) with 10 nmoles of IODO-GEN™ was rinsed with 200 μL of phosphate buffer (0.2 M, pH 8.0) containing sodium chloride (0.15 M). The fresh PBS cocktail (200 μL), Na[125I] (2 μL, 2 mCi, Izotop Hungary, I-RB-41) and aqueous solution of IGF2 (30 μL, 6 μg) were added into the reaction tube. After 5 min of vigorous shaking at room temperature, the 12.5 mM sodium phosphate buffer containing BSA (1 mg/mL) was added to suppress the stickiness of the IGF2 peptide to the reaction tube walls, and the reaction mixture was directly injected on the HPLC system. The desired [125I]-monoiodotyrosyl-Tyr2-IGF2 was separated from the starting IGF2 and over-iodinated IGF2 on reverse phase in water/acetonitrile gradient, using a BioZen C4 column (5μ, 125 × 4.6 mm) tempered at 25˚C with the use of analytical-preparative radio-HPLC (Waters Alliance e2695, 2995 PDA detector and Empower 3 software for data processing), combined with the radioactivity-HPLC flow detector Ramona Star (Elysia-Raytest). A BSA-containing buffer (200 μL, 12.5 mM sodium phosphate buffer with 50 mM NaCl and 0.5 M glycine, 0.06% BSA, pH 7.4) was added into each plastic tube of fraction collector to suppress the stickiness of the desired iodinated peptide. A specific BSA preparation (e.g. Sigma-Aldrich A6003) void of "IGF-binding-like" proteins, which interfere with these binding assays, should preferably be used [57]. The isolated mono-iodinated ligand was analyzed under the same conditions as described above. The Gamma Counter Wizard 2470 (Perkin Elmer) was used for quantification of γ-ionization. The isolated fractions were collected and appropriate aliquots of 125I-IGF2 solution were placed in LoBind® Eppendorf tubes and evaporated to dryness on CentriVap at room temperature for 4 hours. The position of modification of IGF2 with iodine on Tyr2 was determined by an identical parallel preparation of [127I]-monoiodotyrosyl-Tyr2-IGF2 and its analysis was performed by mass spectrometry after digestion with trypsin (see below).

## Mass spectrometry of radiolabeled 125I-IGF2

The MALDI TOF spectra were measured on UltrafleXtreme™ MALDI TOF/TOF mass spectrometer (Bruker Daltonics, Germany) with 1 kHz smartbeam II laser. At first, the extent of iodination was determined. Samples were prepared by dried droplet method (analysis conditions: sample diluted in 50% acetonitrile with 0.1% TFA was applied on DHB matrix and analyzed in positive linear mode, instrumental setting tuned to 5–20 kDa). The accelerating voltage was set at 25kV. Typically, spectrum was obtained by accumulating 5 000 shots. The position of modification of IGF2 by iodine on Tyr residue 2 was revealed, after reduction and alkylation of the sample, using acetamide and by digestion with trypsin, through LC-MS/MS peptide analysis on the ULTIMATE 3000 RSLCnano (Dionex, Thermo Scientific) coupled to TripleTOFTM 5600 (SCIEX).

## Immobilization of D11 and saturation binding assay with [125I]-monoiodotyrosyl-Tyr2-IGF2

Neutravidin (ThermoFisher, 500 ng/well, 100 nM) was adsorbed to black 96-well plates for 60 minutes (MaxiSorp, ThermoFisher) in 50 mM borate buffer pH 9.5. The wells were blocked with a biotin-free casein buffer (SDT GmbH, Germany) for 2–3 hours. Then, the nickel-charged NTA (tris-nitrolacetic acid) decorated iBodies® (IOCB Tech, Prague, Czech Republic, https://www.uochb.cz/en/iocb-tech) containing covalently bound biotin was attached to the

plate through its biotin tag for 60 minutes. Finally, recombinant D11 (166 ng/well, 100 nM) was added to the wells and immobilized on iBodies® via interaction of its His-tag with nickel-loaded NTA moiety of iBodies® for 2 hours. All incubations and washing steps were performed in TBS (20 mM Tris-HCl, 150 mM NaCl, pH 7.4) with 0.1 % (v/v) Tween 20 for incubations, but with 0.05 % (v/v) Tween 20 in the case of washing steps.

For saturation binding experiments, D11 immobilized in wells was incubated in a total volume of 100 μl of a binding buffer (100 mM HEPES pH 7.6, 100 mM NaCl, 5 mM KCl, 1.3 mM MgSO$_4$, 10 mM glucose, 15 mM sodium acetate and 1 % bovine serum albumine) with various concentrations (0–5 nM) of [$^{125}$I]-monoiodotyrosyl-Tyr2-IGF2 (2200 Ci/mmol) for 16 h at 5˚C. Thereafter, the wells were washed twice with the TBS buffer with 0.5 % Tween 20. The bound proteins in the wells were solubilized twice with 300 μl of 0.1 M NaOH that was collected. Bound radioactivity was determined in the γ-counter. Nonspecific binding was determined by measuring the remaining bound radiotracer in the presence of 10 μM unlabeled IGF2 for each tracer concentration. The experiment was performed three times and the results were evaluated in GraphPrism 8, using non-linear regression considering binding to one site.

## Competition radioligand binding assay with non-labeled ligands, [$^{125}$I]-monoiodotyrosyl-Tyr2-IGF2, and immobilized D11

The wells with immobilized D11 were washed with the binding buffer (100 mM HEPES pH 7.6, 100 mM NaCl, 5 mM KCl, 1.3 mM MgSO$_4$, 10 mM glucose, 15 mM sodium acetate and 1 % bovine serum albumine). Thereafter, the immobilized D11 was incubated with increasing concentrations of IGF2 or analog ($10^{-12}$–$10^{-6}$ M) and [$^{125}$I]-monoiodotyrosyl-Tyr2-IGF2 (43000 cpm, 2200 Ci/mmol). The plate was slowly stirred for 16 h at 5 ˚ C in the binding buffer (total volume 100 μl). After the incubation, the wells were washed twice with TBS without Tween and the proteins attached to the wells were solubilized with 0.1 M NaOH. The solubilized solutions from the wells were counted for associated radioactivity. Each point was determined in duplicate. Binding data were analyzed by GraphPad Prism 8 software (GraphPad Software, Inc., San Diego, CA, United States), using the method of non-linear regression and a one-site fitting program considering the potential depletion of free ligand. The dissociation constant of human [$^{125}$I]-monoiodotyrosyl-Tyr2-IGF2 for immobilized D11 was set to 2 nM (see Results).

## Binding affinities of the hormones for the IGF-1 and insulin receptors in membranes of intact cells

The binding affinities of analogs were determined with receptors in the intact cells. Specifically, binding affinities for IGF-1R were determined with mouse fibroblasts transfected with human IGF-1R and with deleted mouse IGF-1R, according to Hexnerova et al. [55]. The [$^{125}$I]-monoiodotyrosyl-IGF1 was used as a radiotracer. Binding affinities for IR-A were determined with human IR-A in human IM-9 lymphocytes as described previously [58]. The [$^{125}$I]-monoiodotyrosyl-TyrA14-insulin was used as a radiotracer. The binding curve of each analog was determined in duplicate and the final dissociation constant ($K_d$) was calculated from at least three (n≥3) binding curves (each curve giving a single $K_d$ value), determined independently and compared to binding curves for human insulin or human IGF-1, depending on the type of receptor.

## Receptor phosphorylation assay

Cell stimulation and detection of receptor phosphorylation were performed as described previously [44], using mouse fibroblasts transfected with human IR-A or human IGF1R. Briefly, the

cells were stimulated with 10 nM concentrations of the ligands for 10 min. Proteins were routinely analyzed using immunoblotting. The membranes were probed with anti-phospho-IGF-1Rβ (Tyr1135/1136)/IRβ (Tyr1150/1151) (Cell Signaling Technology). Each experiment was repeated four times. The data were expressed as the contribution of phosphorylation relative to the human insulin (IR-A) or IGF-1 (IGF1R) signal in the same experiment. Mean ± S.D. (n≥4) values were calculated. The significance of the changes in stimulation of phosphorylation in relation to IGF2 was calculated, using one-way analysis of variance (ANOVA), with Dunnett's test comparing all analogs versus control i.e. IGF2 [44].

## Results

### Production of Leu19-IGF2 analog

Using a recombinant expression in *E. coli*, we prepared Leu19-IGF2 with more than 98 % HPLC purity and in a sufficient quantity (typical yield from 1 L of media was about 400 μg of the pure protein) for performing receptor binding experiments.

### Production of D11

Recombinant expression of His-tagged D11 protein in BL21(λDE3) *E. coli* cells typically resulted in ~8 mg of the pure D11 protein per 1 L of media. The analytical RP-HPLC chromatogram and MS spectrum of purified D11 domain are shown in S1 and S2 Figs in S1 File.

### Preparation of [$^{125}$I]-monoiodotyrosyl-Tyr2-IGF2

Iodination and purification procedures with IGF-2 typically yielded about 120 μCi of HPLC pure [$^{125}$I]-monoiodotyrosyl-Tyr2-IGF2 (6% RCY, >99% RCP) with a specific activity 2200 Ci/mmol. The chromatogram showing purification of a crude reaction mixture after radiolabeling of human IGF-2 is shown in S3 Fig in S1 File. The radiochemical purity of the isolated fraction $^{125}$I-IGF-2 was proved by analytical radio-HPLC (S4 Fig in S1 File). The position of $^{125}$I on Tyr2 in the IGF-2 molecule was localized by mass spectrometry after trypsinolysis.

### Saturation radioligand binding assay on immobilized D11

To determine the dissociation constant ($K_d$) of [$^{125}$I]-monoiodotyrosyl-Tyr2-IGF2 for immobilized D11, we performed saturation binding experiments, where bound radioactivity at increasing concentrations of the radioligand is measured at equilibrium. A representative saturation binding curve is shown in Fig 2. Statistical analysis of the results from three such independent experiments provided the final $K_d$ value 2.0 ± 1.0 nM (n = 3).

### Competition radioligand binding assays on immobilized D11

The binding affinities of human IGF2, IGF1 and Leu19-IGF2 for the D11 domain of IGF2R were tested in a newly developed competition assay using immobilized D11, a fixed concentration of [$^{125}$I]-monoiodotyrosyl-Tyr2-IGF2 (43 000 cpm, 0.02 nM) and various concentrations of non-labeled ligand. Typical total binding was between 4 000–6 000 cpm and nonspecific binding (in an excess of unlabeled IGF2) was typically between 2–5% of the total binding. The representative curves are shown in Fig 3. The $K_d$ of human IGF2 toward the immobilized D11 was 1.0 ± 0.7 (n = 12), but the $K_d$ value of Leu19-IGF2 was 0.28 ± 0.1 (n = 6) nM, showing that this analog is almost four-fold stronger in D11 binding than human IGF2. We observed no inhibition of [$^{125}$I]-monoiodotyrosyl-Tyr2-IGF2 binding to D11 by human IGF-1 in two independent experiments (Table 1).

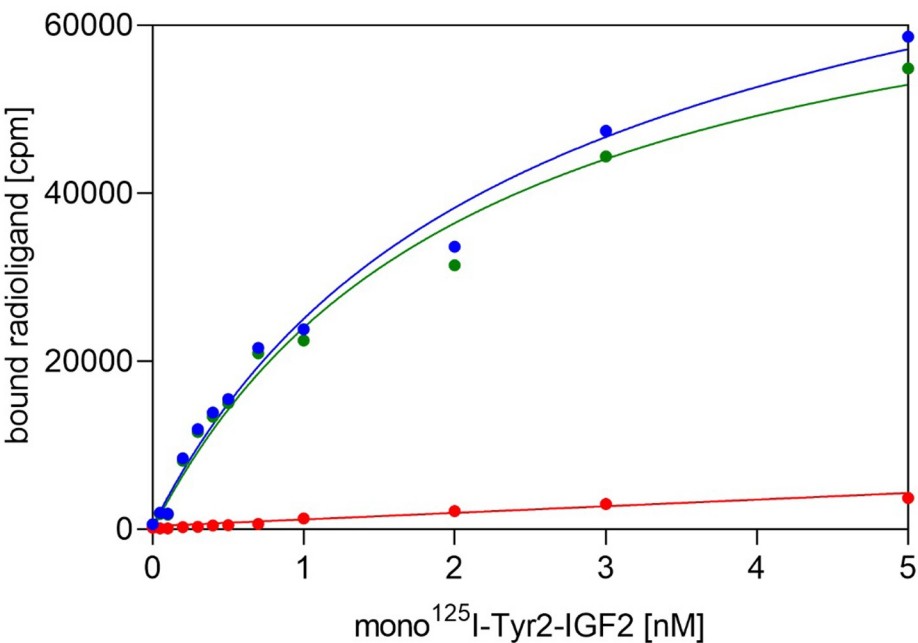

**Fig 2. A typical saturation binding curve of [125I]-monoiodotyrosyl-Tyr2-IGF2 to immobilized domain D11 of IGF2R.** Total binding is in blue, non-specific binding is in red and specific binding is shown in green.

## Binding of IGF hormones to native receptors for insulin and IGF-1 in membranes of intact cells

In order to further validate the results of our new radioligand binding assay for IGF2 receptor and to put them into the context of other receptors for insulin-like hormones, we also

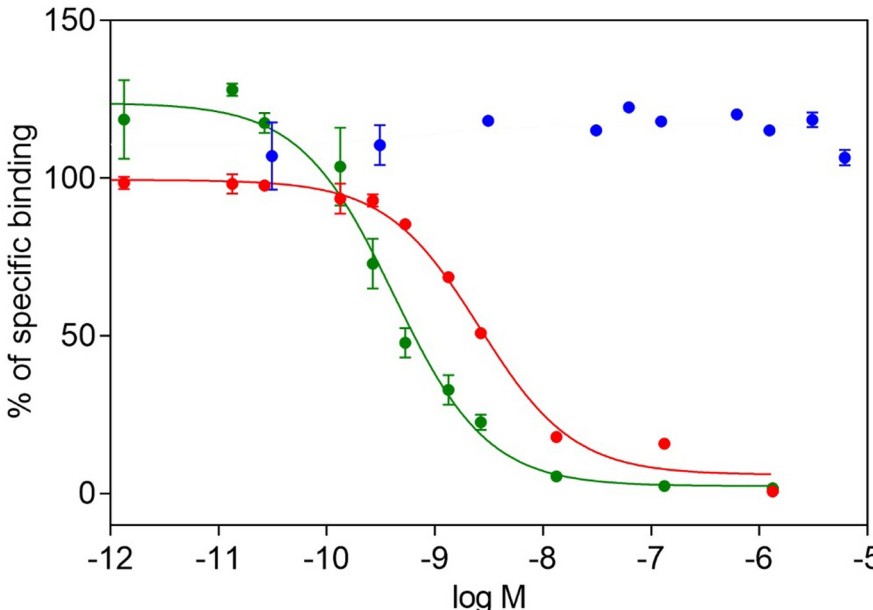

**Fig 3. Representative binding curves of IGF hormones on immobilized D11.** Inhibition of binding of human [125I]-monoiodotyrosyl-Tyr2-IGF2 to D11 by human IGF2 (in red), by human IGF1 (in blue), and by Leu19-IGF2 analog (in green).

**Table 1. Binding affinities of human IGF2, Leu19-IGF2, human insulin and human IGF1 for D11, IR-A and IGF1R (details are provided in methods).**

| Analog | D11 of IGF2R | | IGF1R | | IR-A | |
|---|---|---|---|---|---|---|
| | $K_d$ ± S.D. [nM] (n) | Relative binding affinity [%] | $K_d$ ± S.D. [nM] (n) | Relative binding affinity [%] | $K_d$ ± S.D. [nM] (n) | Relative binding affinity [%] |
| IGF2 | 1.0 ± 0.7 (12) | 100 | 1.6 ± 0.3 (4) | 12.1 | 2.6 ± 0.4 (3) | 10.4 |
| Leu19-IGF2 | 0.28 ± 0.1 (6) | 357 | 1.1 ± 0.1 (2) | 16.7 | 7.3 ± 0.8 (3) | 3.7 |
| human insulin | n.d. | - | 290* | 0.8* | 0.27 ± 0.06 (4) | 100 |
| IGF1 | no binding (2) | - | 0.19 ± 0.08 (5) | 100 | 24* | 0.9* |

The $K_d$ values and relative binding affinities were calculated from at least three independent measurements. (n) is number of replicates. Relative binding affinity is defined as $K_d$ of native hormone/$K_d$ of analog x 100.

*Data from Ref. [44].

measured the binding affinities of human IGF1, IGF2, insulin and Leu19-IGF2 toward human receptor for IGF-1 (IGF1R) and isoform A of insulin receptor (IR-A). The results are summarized in Table 1 and the binding curves are shown in S5 Fig in S1 File. Substitution of phenylalanine at position 19 for leucine has no effect on binding to IGF1R and Leu19-IGF2 analog is equipotent with native IGF-2. On the other hand, Leu19-IGF2 has an approximately 3 times lower binding affinity towards IR-A, than native IGF2 (7.3 *vs* 2.6 nM, respectively).

## Receptor phosphorylation assays

We were also curious to find how Phe19Leu mutation in IGF2 affects an analog's ability to activate receptors for insulin and IGF-1. Hence, we measured the abilities of Leu19-IGF2, human insulin, IGF-1 and IGF to activate IGF1R and IR-A. The results are shown in S6 Fig in S1 File. In general, the abilities of all ligands to induce phosphorylation of both IR-A and IGF1R followed the trends of their binding affinities for these receptors and were in agreement with our previous data [44, 46, 48]. Leu19-IGF2 mutant exhibited a reduced capability to activate IR-A, compared to that of native IGF2 (S6A Fig in S1 File), which coincides well with its reduced binding. The IGF1R stimulation abilities of Leu19-IGF2 (S6B Fig in S1 File) were comparable with that of IGF2, again closely following the trend of their binding affinities for this receptor.

## Discussion

This study was inspired by the growing evidence of the high importance of the IGF2/IGF2R axis in the brain and its newly discovered functions in neural tissues. The IGF2 hormone, which has in the long term been considered mainly as a mitogenic growth factor with a significant role in the development of several types of cancer, and the IGF2R receptor without any clearly defined intrinsic catalytic/signaling activities, are now recognized as important and interacting players in brain functions, such as in the processes of memory enhancement and consolidation [8, 10]. This evidence is supported by an abundant expression of IGF2 in the adult brain and there is IGF2 the most abundantly expressed among all insulin-like peptides with high levels of mRNA expression in myelin sheets, leptomeninges, choroid plexus, hippocampus and cortex [59, 60]. Furthermore, peripheral IGF2 can cross the blood brain barrier by active transport and thus it can be transported into brain tissue via IGF2R localized in the brain capillaries [61–63]. Interestingly, brain IGF2 receptors (IGF2R) were found to be concentrated mainly in the hippocampus, choroid plexus and meninges, although many other brain regions, such as the neocortex, also express them in high levels [59, 60]. Brain

colocalization of both interacting proteins, IGF2 and IGF2R, indicates their concerted action and functions in neural tissues.

Our effort to develop a new methodology for determination of the binding affinity of IGF2 ligand for IGF2R was also motivated by the unavailability of a sensitive and robust IGF2R binding assay. In the past, partially purified rat placental membrane-derived receptors [64] or IGF2R overexpressing baby hamster kidney cells [51, 52] were used for determination of binding affinity. However, the results of these assays could be affected by the presence of IR-A or IGF1R, which can be expected in the cells used for assays. Another existing assay for determination of ligand affinity toward IGF2R is surface plasmon resonance (SPR). This label-free method is based on the kinetic analysis of the interaction and provides $k_{on}$ and $k_{off}$ rate constants, from which a $K_d$ can be determined [16, 24–26]. The disadvantages of SPR are its relatively high cost, specialized instrumentation, non-trivial evaluation of results and often significantly reduced sensitivity compared to other assays. This is mostly caused by the non-specific immobilization of proteins and a loss of their activities. Besides cell-based assays and SPR, isothermal titration calorimetry (ITC) was also used for $K_d$ determination of the interaction of IGF2 with recombinant D11 of IGF2R [53]. Although this steady-state technique can precisely determine a $K_d$ value (and other parameters like ΔG and ΔS), a limiting factor in ITC is the extensive consumption of a protein sample.

For the above-discussed reasons, we developed a new method for assessing the affinity of ligands towards IGF2R, using the well-defined immobilization of D11, which is the key IGF2-binding element in IGF2R. Of all 15 domains of IGF2R, only D11 directly interacts with IGF2 [65]. This binding process is assisted by the D13 domain that modulates ligand binding and flexibility [66]. The unique property of D11 made it a good candidate for the development of a simple *in vitro* binding assay for evaluation of IGF2R ligands. Another advantage of D11, compared to the native membrane bound IGF2R, is its easier accessibility due to the possibility of a recombinant expression in *E. coli*. The use of a protein larger than D11 would be more complicated due to the use of a eukaryotic expression system. Although the ability to bind to D11 is most important for the first screening of the binding affinity of IGF2 mutants to IGF2R, the use of larger IGF2R constructs (e.g., D11-D13 or larger) might be of interest for further characterization of selected IGF2 analogs.

Our new *in vitro* method, similarly to SPR, is based on immobilization of one of the binding partners on a solid surface. But, in contrast to SPR where IGF2 or IGF2 mutant is usually immobilized and IGF2R is used as an analyte, we used the immobilized D11 of IGF2R. The IGF2 molecule is sensitive to modifications and its immobilization can reduce its binding ability. Hence, our approach offers no restrictions for the use of different IGF2 variants. Moreover, D11 can be prepared with a suitable tag that can be exploited for D11 immobilization.

For immobilization of His-tagged D11 of IGF2R, we chose iBodies®, a modular synthetic antibody mimetic on the basis of a hydrophilic polymer [33] (Fig 1B). The iBodies® are composed of a water-soluble and biocompatible N-(2-hydroxypropyl)methacrylamine (HPMA) copolymer that carries a number of reactive groups, enabling covalent attachment of various ligands [67]. The iBodies® are anchored to the surface of polystyrene wells through high-affinity biotin-neutravidin interactions. The nickel-loaded nitrilotriacetic acid ($Ni^{2+}$-NTA) moiety of iBodies® enables the binding of a target protein containing a polyhistidine tag; His-tagged D11, in our case. According to the high affinity of IGF2 for immobilized D11 (Table 1), the localization of His-tag at the C-terminus of the D11 does not apparently affect the interaction of IGF2 with immobilized D11. This is probably because both the C- and N-termini of D11 are localized on the opposite side of D11 from its IGF2 binding surface (Fig 1A). This spatial segregation makes the IGF-2 binding site of D11 fully accessible for interactions with IGF2 ligands and is not affected by immobilization. The interaction of IGF2 and D11 is mediated by

interaction of IGF2's Phe19, which interacts with D11 amino acids Phe1567, Leu1629, and Tyr1542, then by interaction of IGF2 Leu53 with Lys1631 of D11, and finally IGF2's Thr16 creates contacts with Tyr1606 and Ile1572 of D11 [25, 26].

Another important aspect of our methodology was the successful preparation of a specifically monoiodinated IGF2 at the Tyr2 site. Iodine on the Tyr2 ring does not affect D11 binding, due to its localization close to the hormone's N-terminus (Fig 1A). The use of radioligands in binding studies remains to date the most sensitive and accurate approach to measure binding parameters. Iodine-125 is a convenient, easily available isotope with a friendly half-life (59.49 days) and a high specific activity (~ 2200 Ci.mmol$^{-1}$) and offers various options for incorporation into the peptide/protein.

We performed saturation binding assays and determined the $K_d$ value of [$^{125}$I]-monoiodo-tyrosyl-Tyr2-IGF2 for immobilized D11, which is approx. 2 nM. This high-affinity specific binding, together with a low nonspecific binding (below 10 % of total binding) reflect the correct set-up of the experiment (Fig 2). The $K_d$ of a radioligand is necessary for the subsequent calculation of $K_d$ values for cold ligands measured in competitive binding assays. Using this methodology, we determined the $K_d$ values of native IGF2, Leu19-IGF2 mutant and native IGF1 as representative high or low affinity binders [16]. The higher affinity of Leu19-IGF2 analog (0.28 nM, 357% of native IGF2) confirms the data published by Delaine et al. [16], who found that the analog is about 3 times more potent than native IGF2. It is worth noting that human IGF1 does not bind to D11 up to 10$^{-5}$ M concentration (Fig 3). This finding highlights the remarkable insulin-like hormone selectivity of our new binding assay.

In general, we measured significantly higher affinities of ligands for immobilized D11 than other authors. The previously published binding affinities of IGF2 toward D11 varied between 40–100 nM and were determined by SPR [26, 53]. Such different affinities may be due to different experimental setups and most probably because of different kinds of receptor protein or its immobilization. Indeed, binding studies with native insulin receptors in intact cells [68] or in isolated cell membranes [69, 70] determined much higher affinities (~0.2–0.5 nM and ~0.5–1 nM, respectively) than the binding affinities determined by SPR (73 nM and 166 nM) [71].

Furthermore, to put our data into the context of other receptors for insulin-like hormones, we characterized IGF ligands by "classic" radioligand binding assays with IR-A and IGF1R and for their abilities to activate these receptors. The binding affinities of Leu19-IGF2 and native IGF2 for IGF1R are similar (Table 1) and coincide with the data found by others [16, 72]. The abilities of Leu19-IGF2 and IGF2 to stimulate IGF1R are comparable as well (S6 Fig in S1 File). Ziegler at al. [72] previously reported that the binding affinity of Leu19-IGF2 for IR-A is weaker than the affinity of native IGF2 for this receptor. This agrees with our data, showing that the binding affinity of Leu19-IGF2 toward IR-A is approximately 3 times lower than the affinity of native IGF2 (Table 1), which coincides with the abilities of these hormones to stimulate IR-A phosphorylation (Table 1 and S6B Fig in S1 File).

To summarize, we developed a new robust method to determine the binding affinity of IGF2 or its analogs for the D11 domain of IGF2R. The principle is based on the competition between unlabeled ligand (IGF2 or analog) and radioactive iodine-labeled IGF2 for immobilized D11. The use of $^{125}$I-labeled hormones in receptor binding studies represents the most precise and sensitive method for determination of the affinities or kinetic parameters of receptor binders [40, 47, 73, 74]. Furthermore, the great advantage of this assay is the low consumption of both immobilized protein and ligand. An important condition for the development of a sensitive and robust IGF2R binding assay was the well-defined immobilization of D11 on polystyrene plates, using synthetic iBodies®. Regarding the recently discovered importance of the IGF2:IGF2R interactions in the central nervous system [9, 75], our new method could find

applications in the development of IGF2 analogs, which could be clinically important in the treatment of some neurodegenerative diseases.

## Supporting information

**S1 File.**
(PDF)

## Author Contributions

**Conceptualization:** Jiří Jiráček, Lenka Žáková.

**Data curation:** Pavlo Potalitsyn, Irena Selicharová, Kryštof Sršeň, Jelena Radosavljević, Aleš Marek, Lenka Žáková.

**Formal analysis:** Lenka Žáková.

**Investigation:** Lenka Žáková.

**Methodology:** Pavlo Potalitsyn, Irena Selicharová, Kryštof Sršeň, Jelena Radosavljević, Aleš Marek, Kateřina Nováková, Lenka Žáková.

**Project administration:** Lenka Žáková.

**Supervision:** Jiří Jiráček, Lenka Žáková.

**Validation:** Pavlo Potalitsyn, Irena Selicharová, Jiří Jiráček.

**Writing – original draft:** Jiří Jiráček, Lenka Žáková.

**Writing – review & editing:** Jiří Jiráček, Lenka Žáková.

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
