## [Decision Letter · Decision Letter 0]

30 Jul 2020

PONE-D-20-20311

A radioligand binding assay for the insulin-like growth factor 2 receptor

PLOS ONE

Dear Dr. Zakova,

Thank you for submitting your manuscript to PLOS ONE. After careful consideration, we feel that it has merit but does not fully meet PLOS ONE’s publication criteria as it currently stands. Therefore, we invite you to submit a revised version of the manuscript that addresses the points raised during the review process.

Please, address carefully the concerns of both reviewers in a revised version.

We look forward to receiving your revised manuscript.

Kind regards,

Haim Werner

Academic Editor

PLOS ONE

Journal Requirements:

Reviewers' comments:

Reviewer's Responses to Questions

**Comments to the Author**

1. Is the manuscript technically sound, and do the data support the conclusions?

Reviewer #1: Yes

Reviewer #2: Yes

2. Has the statistical analysis been performed appropriately and rigorously? 

Reviewer #1: Yes

Reviewer #2: Yes

3. Have the authors made all data underlying the findings in their manuscript fully available?

Reviewer #1: Yes

Reviewer #2: Yes

4. Is the manuscript presented in an intelligible fashion and written in standard English?

Reviewer #1: Yes

Reviewer #2: Yes

5. Review Comments to the Author

Reviewer #1: This study develops a sensitive radioligand competition binding assay for IGF2R and determined the binding affinities of IGF2, Leu19-IGF2, IGF1 and insulin. Compared to numerous studies on the IGF1 receptor, relatively less efforts have been devoted to the IGF2 receptor. Given the recent findings on the role of IGF2 in memory consolidation and the possible involvement of the IGF2 receptor in this process, this study is timely. The experiments appeared to be performed competently. I have several minor comments that may help to further improve this MS.

1) The use of D11 is a good idea and its advantages are well explained. However, D11 is only a portion of the receptor, other domains (such as D13) also involved in IGF2-IGF2 receptor binding. I suggest the authors to add some discussions about the limitation of this approach.

2) The assay uses IGF2R D11, which is the binding domain known to bind to IGF2. Yet, in the abstract (lines 27-28), it was stated that this “assay will be helpful for the characterization of new ligands … . If the goal is to find additional and new ligands for this receptor, why use the D11 domain? Please explain or revise.

3) Line 76, … the remaining hormone is present at the cell surface”. Do you mean the remaining of IGf2 receptor?

4) The text is rather long and words. I suggest the authors to shorten the MS.

Reviewer #2: This article describes the development of a new robust radioligand binding assay to determine the binding affinity of IGF2 or its analogs for the D11 domain of the IGF2R.

The role of IGF2 in adult human physiology has been very sparsely studied. The IGF2R is classically considered as a non signaling receptor, but a scavenger receptor limiting circulating IGF2 levels to avoid organ overgrowth and overmitogenicity.

The rationale behind the authors interest is a number of recent data suggesting a potential role of the IGF2/IGF2R axis in brain plasticity, learning and memory consolidation, suggesting a possible target for clinical interventions in the treatment of neurodegenerative diseases.

The assay is based on sound principles and its experimental basis well described. The results show a real improvement compared to ITC or SPR methods. The part concerning insulin and IGF-I receptor binding and phosphorylation assays are a bit out of focus and do not bring new information, but help in putting the new data in context.

Specific comments

The part of the introduction regarding the physiological role of IGF2 (lines 32-42) is somewhat superficial. It mentions that “IGF2 is supposed to be a regulator of cell growth, survival, migration and differentiation”. It completely ignores the fact that IGF2 is the major growth factor in fetal development in rodents and humans. The essential work of Efstratiadis and colleagues is ignored. The following papers at least should be quoted:

Efstratiadis A. Genetics of mouse growth. Int J Dev Biol 42:955-976 (1998).

Eggenschwiller J, Ludwig T, Fischer P, Leighton PA, Tilgman SM and Efstratiadis A. Mouse mutant embryos overexpressing IGF-II exhibit phenotypic features of the Beckwith-Wiedeman and Simpson-Golabi-Behmel syndromes. Genes and Development 11:3128-3142 (1997). This paper shows the importance of the scavenger role of the IGF2R.

Lines 38-39: “The significance of IGF2 is supported by its high levels in the serum of adults”: this is true in humans but not in mice where expression shuts off after birth, probably explaining in part while it has been largely ignored in adult physiology.

Line 42: …”an increased risk of developing various cancers”…: it should also be mentioned that IGF2 plays a major role in non-islet-cell tumor (IGF-2-oma) hypoglycemia (ref. 5).

Line 52: reference 5 should be quoted with reference 7.

Line 54: …”IGF2 also binds to the IGF2 receptor…

Line 57: …also known as the cation-independent mannose-6-phosphate (M6P) receptor”…

Line 76: …”the remaining hormone is present at the cell surface”…: it should be “the remaining receptor…”

Lines 98-99: … “it is less clear whether IGF2R itself is able to initiate transmembrane signaling (33-35)”: the work of the Karolinska group should be cited here, e.g.:

Tally M, Hall K. Insulin-like growth factor II effects mediated through insulin-like growth factor II receptors. Acta Paediatr Scand suppl 367:67-73 (1990).

Line 100: Reference 36 is wrong, it is about preptin and osteocalcin, nothing to do with IGF2 signaling.

Lines 253-262: “Binding affinities of the hormones for the IGF-1 and insulin receptors in membranes of intact cells”: it should be mentioned that IGF-1 and insulin radioiodinated tracers were used (it is only mentioned in legend of Fig. S5).

Line 431-432: …”than for others”… is unclear, it means “than other authors”…

6. PLOS authors have the option to publish the peer review history of their article (what does this mean?). If published, this will include your full peer review and any attached files.

Reviewer #1: No

Reviewer #2: **Yes: **Pierre De Meyts

---

## [Author Response · Author response to Decision Letter 0]

12 Aug 2020

Prague, 11st August 2020

Prof. Haim Werner

Academic Editor

PLOS ONE

Dear Prof. Werner,

We would like to thank you and the reviewers for the careful assessment of our work and detailed comments. These comments were very valuable, helping us to clarify the major points of the paper and to remove ambiguities. We hope that we have addressed all concerns raised by the reviewers and implemented their suggestions effectively. Our point-by-point responses to the reviewers’ comments are as follows. All changes, which we made in the original manuscript are highlighted in blue. 

Reviewer #1: 

1) The use of D11 is a good idea and its advantages are well explained. However, D11 is only a portion of the receptor, other domains (such as D13) also involved in IGF2-IGF2 receptor binding. I suggest the authors to add some discussions about the limitation of this approach.

 We added two sentences (lines 417-421) explaining the disadvantage of use of larger construct of IGF2R. However, the larger construct (e.g. D11-D13) might be useful for future application. 

2) The assay uses IGF2R D11, which is the binding domain known to bind to IGF2. Yet, in the abstract (lines 27-28), it was stated that this “assay will be helpful for the characterization of new ligands … . If the goal is to find additional and new ligands for this receptor, why use the D11 domain? Please explain or revise.

There is inaccuracy, our assay with D11 is specifically designed for the evaluation of IGF2 mutants. We corrected it to “IGF2 mutants”.

3) Line 76, … the remaining hormone is present at the cell surface”. Do you mean the remaining of IGf2 receptor?

 Yes, it means remaining IGF2R, we corrected it. 

4) The text is rather long and words. I suggest the authors to shorten the MS.

 We tried to shorten a bit the manuscript. We removed repetitive and redundant sentences especially in Introduction. 

Reviewer #2: 

Specific comments

Lines 32-42: The part of the introduction regarding the physiological role of IGF2 is somewhat superficial. It mentions that “IGF2 is supposed to be a regulator of cell growth, survival, migration and differentiation”. It completely ignores the fact that IGF2 is the major growth factor in fetal development in rodents and humans. The essential work of Efstratiadis and colleagues is ignored. The following papers at least should be quoted: Efstratiadis A. Genetics of mouse growth. Int J Dev Biol 42:955-976 (1998).

Eggenschwiller J, Ludwig T, Fischer P, Leighton PA, Tilgman SM and Efstratiadis A. Mouse mutant embryos overexpressing IGF-II exhibit phenotypic features of the Beckwith-Wiedeman and Simpson-Golabi-Behmel syndromes. Genes and Development 11:3128-3142 (1997). This paper shows the importance of the scavenger role of the IGF2R.

 We added these references to relevant parts of manuscript.

Lines 38-39: “The significance of IGF2 is supported by its high levels in the serum of adults”: this is true in humans but not in mice where expression shuts off after birth, probably explaining in part while it has been largely ignored in adult physiology.

 We corrected it.

Line 42: …”an increased risk of developing various cancers”…: it should also be mentioned that IGF2 plays a major role in non-islet-cell tumor (IGF-2-oma) hypoglycemia (ref. 5).

 We corrected it.

Line 52: reference 5 should be quoted with reference 7.

 We added the reference.

Line 54: …”IGF2 also binds to the IGF2 receptor…

 We corrected it.

Line 57: …also known as the cation-independent mannose-6-phosphate (M6P) receptor”…

 We corrected it.

Line 76: …”the remaining hormone is present at the cell surface”…: it should be “the remaining receptor…”

 We corrected it.

Lines 98-99: … “it is less clear whether IGF2R itself is able to initiate transmembrane signaling (33-35)”: the work of the Karolinska group should be cited here, e.g.:

 Tally M, Hall K. Insulin-like growth factor II effects mediated through insulin-like growth factor II receptors. Acta Paediatr Scand suppl 367:67-73 (1990).

 We added the reference.

Line 100: Reference 36 is wrong, it is about preptin and osteocalcin, nothing to do with IGF2 signaling.

 The reviewer was right, we corrected the reference. 

Lines 253-262: “Binding affinities of the hormones for the IGF-1 and insulin receptors in membranes of intact cells”: it should be mentioned that IGF-1 and insulin radioiodinated tracers were used (it is only mentioned in legend of Fig. S5).

 We added the information about radioiodinated tracers to Methods section.

Line 431-432: …”than for others”… is unclear, it means “than other authors”…

 We corrected it. 

We hope that the response together with changes and corrections will fulfill the referees’ as well as your requirements, and that this revised manuscript could be considered for publication in the PLOS ONE. Please do not hesitate to contact me in case of any further queries.

Yours sincerely, 

Lenka Žáková

Institute of Organic Chemistry and Biochemistry

Academy of Sciences of the Czech Republic

Flemingovo nám. 2, 166 10 Praha 6

Czech Republic

Phone: +420-2-20183441, Fax: +420-2-20183571

E-mail: zakova@uochb.cas.cz

---

## [Editor Report · Decision Letter 1]

17 Aug 2020

A radioligand binding assay for the insulin-like growth factor 2 receptor

PONE-D-20-20311R1

Dear Dr. Zakova,

We’re pleased to inform you that your manuscript has been judged scientifically suitable for publication and will be formally accepted for publication once it meets all outstanding technical requirements.

Kind regards,

Haim Werner

Academic Editor

PLOS ONE

Additional Editor Comments (optional):

The authors have satisfactorily addressed reviewer's comments.
---

## [Editor Report · Acceptance letter]

21 Aug 2020

PONE-D-20-20311R1 

A radioligand binding assay for the insulin-like growth factor 2 receptor 

Dear Dr. Žáková:

I'm pleased to inform you that your manuscript has been deemed suitable for publication in PLOS ONE. Congratulations! Your manuscript is now with our production department. 

Kind regards, 

on behalf of

Dr. Haim Werner 

Academic Editor

PLOS ONE